# *Lactobacillus rhamnosus* GG Derived Extracellular Vesicles Modulate Gut Microbiota and Attenuate Inflammatory in DSS-Induced Colitis Mice

**DOI:** 10.3390/nu13103319

**Published:** 2021-09-23

**Authors:** Lingjun Tong, Xinyi Zhang, Haining Hao, Qiqi Liu, Zihan Zhou, Xi Liang, Tongjie Liu, Pimin Gong, Lanwei Zhang, Zhengyuan Zhai, Yanling Hao, Huaxi Yi

**Affiliations:** 1College of Food Science and Engineering, Ocean University of China, Qingdao 266003, China; m18754030882@163.com (L.T.); zhang17860815936@163.com (X.Z.); haohaining1994@163.com (H.H.); Liuqiqi9306@126.com (Q.L.); zhouzihan@stu.ouc.edu.cn (Z.Z.); liangxi6029@163.com (X.L.); ltjpeak@126.com (T.L.); gongpimin@outlook.com (P.G.); zhanglanwei@ouc.edu.cn (L.Z.); 2Key Laboratory of Functional Dairy, Co-Constructed by Ministry of Education and Beijing Municipality, College of Food Science and Nutritional Engineering, China Agricultural University, Beijing 100083, China; zhaizy@cau.edu.cn (Z.Z.); haoyl@cau.edu.cn (Y.H.); 3College of Food Science and Nutritional Engineering, China Agricultural University, Beijing 100083, China

**Keywords:** *Lactobacillus rhamnosus* GG, extracellular vesicles, ulcerative colitis, inflammatory, gut microbiota

## Abstract

Ulcerative colitis (UC) is a relapsing and remitting inflammatory disease. Probiotics have a potential beneficial effect on the prevention of UC onset and relapse in clinical trials. *Lactobacillus rhamnosus* GG (*L. rhamnosus* GG) have shown clinical benefits on UC patients, however, the precise mechanisms are unknown. The aim of this study is to explore the effect of extracellular vesicles released from *L. rhamnosus* GG (LGG-EVs) on dextran sulfate sodium (DSS)-induced colitis and propose the underlying mechanism of LGG-EVs for protecting against colitis. The results showed that LGG-EVs could prevent colonic tissue damage and shortening of the colon (*p* < 0.01), and ameliorate intestinal inflammation by inhibiting TLR4-NF-κB-NLRP3 axis activation. Consistently, the pro-inflammatory cytokines (TNF-α, IL-1β, IL-6, IL-2) were suppressed effectively upon LGG-EVs treatment (*p* < 0.05). The 16S rRNA sequencing showed that LGG-EVs administration could reshape the gut microbiota in DSS-induced colitis mice, which further alters the metabolism pathways of gut microbiota. These findings propose a novel perspective of *L. rhamnosus* GG in attenuating inflammation mediated by extracellular vesicles and offer consideration for developing oral gavage of LGG-EVs for colitis therapies.

## 1. Introduction

Ulcerative colitis (UC) is an idiopathic inflammatory bowel disorder of the colon that causes dysregulation of gut microbiota and continuous mucosal inflammation. UC exhibits a higher prevalence worldwide, reaching 7.6–245 cases per 100,000 persons/year [1]. Genetics, environmental factors, autoimmunity and gut microbiota were considered to increase the risk of UC [2]. Given the complexity and appearing complications induced by immunosuppressive drugs, it is significant to developing novel therapeutic approaches and alternative drugs to treat UC.

Some probiotics, as nutritional supplements in functional foods, have been shown to promote intestinal health by enhancing epithelial function and reshaping gut microbial homeostasis. It was reported that *Bifidobacterium infantis* 35624 could exert beneficial immunoregulatory effects in the mucosal immune system [3]. *Lactobacillus rhamnosus* GG (*L. rhamnosus* GG, ATCC 53103) could enhance intestinal functional maturation and IgA production and protect against colitis [4]. *Lactobacillus delbrueckii* ssp *bulgaricus* (LOT No. FK0201, LB-G040) attenuated the clinical signs of intestinal inflammation inducing a decrease in inflammatory cytokines [5]. In addition, prebiotics might change the composition of gut microbiota, improve the function of the intestinal barrier, enhance intestinal immunity. It was shown that prebiotic fructans and resveratrol treatment could increase the amount of *Bifidobactrium* and *Lactobacillus* in DSS-induced colitis [6]. Therefore, prebiotics and probiotics may represent a valid armamentarium to alleviate colitis, while the mechanism of the action is still unclear. In 2006, a clinical trial had been documented that *L. rhamnosus* GG could be effective and safe for maintaining remission in patients with ulcerative colitis [7]. However, the definite mechanism of *L. rhamnosus* GG initiating anti-inflammatory activity remains incompletely illuminated.

The latest study showed that extracellular vesicles (EVs) played a crucial role in bacteria–bacteria and bacteria–host interactions [8]. EVs derived from bacteria are spherical and nano-sized vesicles, and the diameter ranges from 20 to 400 nm. Like mammalian EVs, probiotic-derived EVs carry a large amount of microRNA, mRNA, proteins, and other active factors, which can mediate communications of cell–microbiota–host [9]. It was speculated that EVs derived by bacteria played a key role in physiological and pathological functions by transferring cargoes to recipient cells. The beneficial immunomodulatory effects of probiotic-derived EVs were studied based on in vivo or in vitro assays [10]. *A. muciniphila* (ATCC BAA-835) derived EV was reported as a beneficial factor that ameliorated the production of pro-inflammatory IL-6 from colon epithelial cells [11]. EVs derived from *Lactobacillus kefirgranum* PRCC-1301 might have an anti-inflammatory effect on colitis by inhibiting the NF-κB pathway and improving intestinal barrier function [12]. However, many enigmatic puzzles regarding the underlying functions of Gram-positive EVs still remain unclear.

Given the efficacy of *L. rhamnosus* GG in patients with ulcerative colitis, whether the EVs released from *L. rhamnosus* GG are involved in anti-inflammatory response and the mechanism remain unknown. In this study, the aim was to investigate the effect of *L. rhamnosus* GG-EVs (LGG-EVs) on anti-inflammatory properties based on the TLRs-NF-κB-NLRP3 signaling pathway and gut microbiota in DSS-induced colitis mice.

## 2. Materials and Methods

### 2.1. Preparation of L. rhamnosus GG Derived EVs

*L. rhamnosus* GG was purchased from ATCC (53103) and cultured in MRS medium at 37 °C for 24 h. Then medium was centrifuged at 8000 × *g* for 30 min at 4 °C to remove cells and other debris. The supernatant was filtered by 0.22 μm PVDF filter and concentrated using Centricon™ Plus-70 Centrifugal Filter Units (MilliporeSigma™). The filtered suspension was ultracentrifuged at 100,000 g using HIMAC CP70ME Ultracentrifuge (Hitachi, Ltd., Tokyo, Japan) at 4 °C for 2 h. The pellets were resuspended with PBS, and washed by ultracentrifugation at 100,000 × *g* for 60 min at 4 °C. LGG-EVs were collected and stored at −80 °C. The size distribution of LGG-EVs was measured by Dynamic Light Scattering (DLS), and all samples were evaluated in three replicates. The morphology of LGG-EVs was visualized via Transmission Electron Microscope (TEM). The morphology was imaged using HITACHI H-7650 Transmission Electron Microscope (Hitachi, Ltd., Tokyo, Japan).

### 2.2. Animals and Treatments

Specific pathogen-free (SPF) C57BL/6J male mice (4–5 weeks of age) were purchased from the Laboratory Animal Breeding Center of Pengyue (Jinan, China). Mice were maintained at 22 °C with 12-h light/dark cycles and kept in individual cages. A standard diet (Appendix A) and water were provided ad libitum. The relative humidity was 30% to 70% in the experimental room. Bedding-change and cage-washing was carried out frequently and the preparation of recycled air was used to reduce the stress associated with the experiment. After adaptation for 1 week, mice were randomly divided three groups, including the control group, DSS group (3.5% DSS), and treatment group (3.5% DSS + 1.2 mg/kg LGG-EVs of body weight). Specifically, LGG-EVs were pre-administered to mice for 2 weeks by daily gavage of LGG-EVs (1.2 mg/kg of body weight) according to the latest study [13]. Control and DSS group mice were fed an equal volume of PBS. Then 3.5% DSS (Yeasen Biotech, Shanghai, China) were added into the water for inducing colitis. LGG-EVs were administered until the end of the experiment. The body weight and presence of blood in the stool were used to monitor the development of colitis. All experimental processes were approved by the Committee on the Ethics of Animal Experiments of Ocean University of China (permission number: spxy20200820215).

### 2.3. Western Blot Analysis

Total proteins were extracted from colonic tissues and LGG-EVs using RIPA buffer containing protease and phosphatase inhibitor (Beyotime Biotechnology, Shanghai, China). The equal quantity of protein was loaded into gels, and then transferred onto PVDF membranes (Thermo Fisher Scientific). Proteins were quantified by Image J software (NIH, Bethesda, MD). Antibodies against TSG101 (ab225877), p65 (ab16502), p-p65 (ab76302), NLRP3 (ab210491), ASC (ab180799) were purchased from Abcam (Cambridge, UK). β-actin (GB12001) was purchased from Servicebio (Wuhan servicebio technology CO., LTD, Wuhan, China).

### 2.4. Quantitative Real-Time PCR

Colonic tissues were employed to evaluate the relative expression of inflammatory cytokines. Total RNA was extracted using Trizol Reagent (TIANGEN Biotech, Beijing, China), and 1 μg of total RNA was reverse transcribed using High-Capacity cDNA Reverse Transcription Kit (Applied Biosystems, Waltham, MA, USA). The expression of IL-6, TNF-α, IL-1β, IL-2, TLR-4, Myd88, and GAPDH genes were quantitated using Power Up SYBR Green MasterMix on a StepOnePlus Real-Time PCR Instrument (Applied Biosystems). GAPDH was used as a housing gene. Primers were included in Appendix A.

### 2.5. Enzyme-Linked Immunosorbent Assay

The concentrations of TNF-α, IL-6, and IL-1β were determined in serum using ELISA kits according to the manufacturer’s instructions (Nanjing Jiancheng Bio, Nanjing, China). In brief, serum was obtained by centrifugation at 5000× *g* for 30 min at 4 °C. The absorbance was detected at 450 nm by Multiskan FC (Thermo Scientific, Waltham, MA, USA).

### 2.6. Tissue Histology

The colonic segments were taken and fixed using 4% paraformaldehyde. Then the paraffin sections of each colon were stained with haematoxylin and eosin (H&E). The colonic thin sections were observed under a light microscope (OLYMPUS) at a magnification of 100×.

### 2.7. Microbiota 16S rRNA Gene Sequencing

Total genomic DNA was extracted from the fecal samples of three groups. The bacterial 16S rRNA genes V3–V4 region was amplified using the forward primer 338F (5′-ACTCCTACGGGAGGCAGCA-3′) and the reverse primer 806R (5′-TCGGACTACHVGGGTWTCTAAT-3′). Then the amplicons were pooled in equal amounts, and pair-end 2 × 300 bp sequencing was performed using the Illlumina MiSeq platform with MiSeq Reagent Kit V3 at Shanghai Personal Biotechnology Co., Ltd. (Shanghai, China). High-quality sequences were clustered into OTUs at 97% sequence identity by UCLUST. Microbial functions were predicted by PICRUSt according to high-quality sequences. LEfSe was used to investigate bacterial members that drive differences between groups.

### 2.8. Statistical Analysis

Data were presented as the mean ± standard deviation (SD) or the median, min and max values according to the normality of distribution. One-way analysis of variance (ANOVA), Duncan test and Normality and Lognormality Tests (D’Agostino and Pearson test) was performed using SPSS 22 (SPSS Inc., Chicago, IL, USA) and GraphPad Prime 8 (GraphPad Software, San Diego, CA, USA). All other statistical tests were performed using the GraphPad Prime 8. Statistical differences were considered significant at *p* < 0.05.

## 3. Results

### 3.1. Characterization of L. rhamnosus GG-Derived EVs

To obtain high-yield *L. rhamnosus GG* derived extracellular vesicles (LGG-EVs), *L. rhamnosus GG* culture medium was concentrated firstly using ultrafilter, and then isolated by ultracentrifugation. The protein concentration of LGG-EVs reached 478.43 ± 25.34 μg/mL (Figure 1A). The isolated LGG-EVs were characterized by dynamic light scattering (DLS), western blot (WB), and transmission electron microscopy (TEM). The mean diameter of LGG-EVs was 161.9 ± 54.8 nm, which was consistent with the size distribution of extracellular vesicles defined by MISEV2018 [14]. EVs isolated from the *L. rhamnosus GG* were characterized by WB for the commonly expressed exosomal proteins TSG101 (Figure 1A), and TEM images also exhibited a nano-sized vesicle with a lipid bilayer (Figure 1B).

### 3.2. L. rhamnosus GG-Derived EVs Alleviated DSS-Induced Colitis

A number of studies showed that *L. rhamnosus* GG had remarkable effects on inflammation and infection [15,16]. Whether LGG-EVs have anti-inflammatory effects on colitis remains unknown. In this study, the mouse experimental colitis model was used for verifying the immunomodulatory effect of LGG-EVs. Mice were gavaged with LGG-EVs for 14 days before 3.5% DSS was provided in drinking water. A significant decrease in body weight was observed in DSS-treated mice, while LGG-EVs slightly reversed the body weight loss from 6 to 8 days (Table 1). The spleen was one of the main immune organs, and its weight would increase with the acute inflammatory process [17]. DSS induced the increase in spleen index significantly, whereas LGG-EVs treatment prevented DSS-induced spleen index incensement (Table 2). The liver index showed no significant difference among the three groups (Table 2). LGG-EVs treated mice also reduced significantly less colonic shortening (Table 2, Figure 2A) and inflammatory infiltrate in the mucosa (Figure 2B) when compared to DSS-induced mice. Cytokines, as a crucial regulator in the pathogenesis of colitis, could drive intestinal inflammation and associated symptoms [18]. We therefore evaluated the dysregulated cytokines profile in DSS-induced colitis with or without LGG-EVs treatment. The DSS-induced mice exhibited higher levels of pro-inflammatory cytokines in the colonic tissues and serum compared with healthy mice. In contrast, LGG-EVs treatment suppressed the over-expression of DSS-induced TNF-α, IL-1β, IL-6, and IL-2 at protein and gene level (Table 3, Appendix A). Mechanistically, the expression of TLR4, Myd88 genes and p65, p-p65 proteins in the NF-κB signaling pathway and the triggers activation of the Nod-like receptor family (ASC, NLRP3 proteins) in NLRP3 signaling pathway were investigated, which mainly leads to the synthesis of pro-inflammatory cytokines [19,20]. As displayed in Figure 2C and Table 4, TLR4, Myd88, p65, p-p65 and ASC, NLRP3 proteins were activated by DSS, while LGG-EVs treatment reversed their overexpression. Notably, a previous study reported that the inhibitory effects of *L. rhamnosus* GG on inflammatory mediator expression through TLR4/MyD88/NF-κB signaling pathway [21]. Therefore, LGG-EVs might play a key role in the DSS-induced inflammatory response through its down-regulation of the TLR4-MyD88 axis.

### 3.3. L. rhamnosus GG-Derived EVs Regulated Gut Microbiota in Colitis Mice

Multiple studies have demonstrated that the key role of gut microbiota in the pathogenesis of colitis [2]. The disorder of gut microbiota would break intestinal homeostasis and stimulate the production of pro-inflammatory cytokines. In contrast, the restoration of dysregulated flora could alleviate the imbalance of intestinal immune homeostasis, and thereby inhibit the inflammatory response in the intestine [22]. Hereby, the microbial disorder and characteristics in DSS-induced colitis were confirmed. In a line with other studies, the bacterial richness and α-diversity significantly decreased in DSS-induced mice compared to the control group, which was evident by the change of Chao 1, Faith_pd, Shannon and Simpson indexes, etc. (Table 5). Furthermore, altered microbial construction (β-diversity) was examined by the principal coordinate analysis (PCoA) plot, which clarified changes in the overall structure of gut microbiota. As shown in Figure 3, PCoA revealed significant separation in the community composition based on the OTU level. Notably, LGG-EVs treatment increased significantly the α-diversity of gut microbiota (Table 5) and thereby reversed the dysregulated microbiota community structure of DSS-induced colitis (Figure 3).

To further explore the phenotypic changes of the taxonomic composition in DSS-induced colitis mice with or without LGG-EVs treatment, the taxonomy-based analysis and hierarchical clustering analysis were performed to identify the differences of bacterial abundance at different levels. We first evaluated the effect of LGG-EVs on the abundance of phyla and genera in DSS-treated mice. The taxonomic analysis showed that *Firmicutes* and *Bacteroidetes* were the most abundant phyla, and the relative abundance and ratio of these two phyla were significantly affected by DSS treatment. However, compared with DSS-induced colitis mouse, mice gavaged LGG-EVs had higher relative abundances of these two phyla and lower *Proteobacteria* and *Epsilonbacteraeota* (Figure 4A). Furtherly, LGG-EVs treatment decreased the ratio of *Firmicutes* to *Bacteroidetes* (*F/B*) in the colonic microbiome (Figure 4B). Generally, a low *F/B* ratio contributed to a healthy condition [23]. At the genus/species level, the proportions of certain bacterial groups were impacted by DSS-induced. *Muribaculaceae*, *Rikenellaceae_RC9_gut_group*, and *Lachnospiraceae_NK4A136_group* were decreased, whereas *Helicobacter* and *Escherichia–Shigella* were significantly increased in the DSS-induced colitis mice (Figure 4C). Consistently, DSS resulted in a significant increase in *Helicobacter_ganmani* at the species level (Appendix A), which was closely associated with inflammatory diseases induced by harmful pathogens [24]. Moreover, within the phylum of *Proteobacteria*, the taxonomic branches of *Proteobacteria–Enterobacteriales–Enterobacteriaceae–Enterobacter* and *Escherichia–Shigella* were significantly increased in colitis mice (Appendix A). Obviously, the relative abundance of these harmful pathogens was decreased in LGG-EVs treated mice, which indicated that LGG-EVs could effectively reverse the flora dysregulation of DSS-induced to some extent. Furthermore, the hierarchical clustering analysis demonstrated that the DSS group was different from the control and LGG-EVs group based on the unweighted UniFrac distance (Figure 4D). Overall, oral gavage of LGG-EVs could increase bacterial α-diversity and restore the taxonomic imbalance of gut microbiota induced by DSS.

It had been reported the changes of characteristic bacteria associated with the intestinal infection after *L. rhamnosus* GG treatment [25]. The effect of LGG-EVs on the characteristic bacteria in DSS-induced colitis was explored in this study. As we expected, LGG-EVs treatment increased significantly the relative abundance of *Lachnospiraceae*, *Ruminiclostridium_9*, *Clostridiales_vadinBB60_group*, and *Faecalibaculum* at the family (Table 6) and genus levels (Table 7) in DSS-induced colitis, which was consistent with the greater abundance of *Lachnospiraceae* and *Ruminococcaceae* reported in *L. rhamnosus* GG-treated mice [26]. At the species level, the clustering analysis indicated that both of LGG-EVs and the control group had similar characteristics of the gut microbiota compared to the DSS group (Figure 5A). It might be owing to the difference of some characteristic bacteria associated with colitis, such as the significant increase in harmful bacteria (*Helicobacter_ganmani*, *Clostridium_perfringens*, etc.) in DSS-induced colitis, and decrease in beneficial bacteria (*Bifidobacterium_animalis*, *Clostridiales_bacterium*, etc.). The significant differences at the species level were also observed in both DSS and DSS + LGG-EVs groups based on cluster heatmap analysis (Figure 5B). Obviously, the greater abundance of harmful bacteria was clustered in DSS-induced colitis compared to LGG-EVs treated mice (*p* < 0.05). In contrast, the characteristic beneficial bacteria were increased significantly and clustered in the LGG-EVs group, including *Akkermansia_muciniphila*, *Bifidobacterium_animalis*, etc. (*p* < 0.05, Figure 5B). To further explore the potential microbial markers associated with colitis before and after LGG-EVs treatment, a model that implements a 0-inflated Gaussian distribution of mean group abundance for each taxa (metagenomeSeq) were used. As displayed in Figure 5C, the characteristic bacteria were presented in DSS-induced colitis compared to healthy mice, including *Helicobacter*, *Odoribacter*, *Desulfovibrio*, etc. Meanwhile, *Odoribacter*, *Alistipes*, *Muribaculaceae*, *Lachnospiraceae_NK4A136_group*, and *Akkermansia*, etc. as main characteristic bacteria were enriched in LGG-EVs treated mice compared with DSS-induced colitis mice (Figure 5D). Interestingly, we found that the abundance of *Odoribacter* was increased in DSS-induced colitis, however, the much greater abundance of *Odoribacter* were enriched in LGG-EVs treated mice compared to DSS-induced colitis (Red box, Figure 5C,D). The convincing evidence showed that *Odoribacter* could be considered as a potential marker of some diseases, such as an abdominal abscess in humans [27]. More often, however, their presence is shown to be beneficial for the prevention of several diseases such as IBD [28]. Therefore, *Odoribacter* might be a characteristic bacteria associated with colitis after LGG-EVs treatment.

To further elucidate the effect of LGG-EVs on the functional alterations of gut microbiota, the Kyoto Encyclopedia of Genes and Genome (KEGG) pathway analyses were used among the three groups. As shown in Figure 6A, fluorobenzoate degradation, a closely correlated process with the severity of intestinal inflammation [29], was identified as a significant positive correlation KEGG pathway in DSS-induced colitis compared with healthy mice (*p* < 0.01). Instead, it was identified as a significant negative correlation KEGG pathway in LGG-EVs treated mice (Figure 6B, *p* < 0.05). Consistently, gut microbiota involved in the shigellosis pathway was more abundant in colitis (*p* < 0.05). However, LGG-EVs treatment reversed this change significantly (*p* < 0.01). In addition, the valine, leucine and isoleucine degradation pathway were identified as significant functional KEGG pathways in LGG-EVs treated mice (*p* < 0.05, Figure 6B), which was closely positively correlated with *Akkermansia* [30]. Consistently, the relative abundance of *Akkermansia* was significantly increased in LGG-EVs treated mice (Figure 5B). Thus, the increased level of valine, leucine and isoleucine degradation pathway, and decreased level of both fluorobenzoate degradation and shigellosis pathway after LGG-EVs treatment might contribute to an improvement in DSS-induced colitis.

### 3.4. Correlations among LGG-EVs, Cytokines and Gut Microbiota

To explore the possible relationships among LGG-EVs, gut microbiota and pro-inflammatory cytokines, Spearman’s correlation analyses were performed. As shown in Figure 7A, eight bacteria were significantly negatively correlated with pro-inflammatory cytokines (Table 3), including *Muribaculaceae*, *Rikenellaceae_RC9_gut_group*, *Akkermansia, Eubacterium_fissicatena_group*, *Faecalibaculum, Dubosiella*, *Alistipes*, and *Lactobcillus*. Notably, seven of eight bacteria enriched in the LGG-EVs group compared to DSS-induced colitis (Figure 6). Furthermore, three of four bacteria enriched in the DSS group, including *Helecobacter*, *Clostridium_sensu_stricto_1*, *Oscillibacter*, which were significantly positively correlated with IL-6, IL-1β, TNF-α, TLR4, etc. (*p* < 0.05). To illustrate the value of correlation analyses, a random forest classifier model was constructed that could specifically identify the importance of characteristic bacteria. As displayed in Figure 7B, the top 20 characteristic bacteria were presented. However, six bacteria were related to cytokines (Figure 7A). Among them, four bacteria had a significantly negative correlation with pro-inflammatory cytokines, such as *Muribaculaceae*, *Dubosiella*, *Rikenellaceae_RC9_gut_group*, and *Alistipes*. These findings indicated that these four bacteria might be useful targets for distinguishing the changes with or without LGG-EVs treatment.

## 4. Discussion

In this study, the effect of *L. rhamnosus GG*-derived extracellular vesicles (LGG-EVs) on the DSS-induced colitis was investigated in the mice model. To date, *L. rhamnosus* GG was used for clinical study in many gastrointestinal diseases (NCT01790035, NCT01773967). However, there is no uniform understanding of the mechanism of its probiotic function. The role of Gram-positive EVs on health and disease has been drawing more attention recently. Accumulating evidence has indicated that the probiotic effects of some probiotic bacteria may not depend on the bacteria being alive but on the metabolites released by probiotic bacteria, such as EVs or carried effector molecules in EVs [31,32]. The latest study demonstrated that the function of *L. rhamnosus* GG was mediated through its nanoparticles (EVs), which protected against alcohol-associated liver disease through intestinal aryl hydrocarbon receptor [13]. To our knowledge, the immunomodulatory effect of LGG-EVs on experimental colitis remains unknown. Hereby, a mouse colitis model was induced by DSS. Consistent with the treatment of *L. rhamnosus* GG in the previous report [7], LGG-EVs seems to be effective and safe for maintaining remission in colitis mouse. In addition, a single dose of LGG-EVs was used in this study, more doses should be chosen for comprehensively clarifying the anti-inflammatory effect of LGG-EVs in subsequent research.

It is confirmed that TLR4 signaling is critical for colon carcinogenesis in colitis, and the inhibition of TLR4 signaling may be useful for protecting against colitis or colitis-associated cancer [33]. NF-κB, the key regulator in the mucosal immune system, could be activated by upstream stimuli TLR4. Therefore, inhibiting TLR4 signaling can effectively regulate the expression levels of inflammatory cytokines induced by DSS. It had been reported that *L. rhamnosus* GG or *L. rhamnosus* GG-derived factors could attenuate the inflammatory response induced by activation of TLR4/NF-κB in the colitis [34,35]. In our current research, we found LGG-EVs had the same anti-inflammatory effect, which was accomplished by downregulating the expression of myd88, p65, and p-p65 via inhibiting TLR4 signaling. Accordingly, the expression of pro-inflammatory cytokines at the gene and protein levels were decreased in DSS-induced colitis mice treated with LGG-EVs. NLRP3, the member of the NLRs family, is rapidly emerging as a crucial regulator of the intestinal homeostasis, which could induce production of IL-1β and facilitate the activation of NF-κB signaling cascades, and then initiate the immune response [36]. Interestingly, Toll-like receptors (TLRs) could induce the synthesis of mitochondrial DNA, which is crucial for NLRP3 signaling. Taken together, a closely TLRs-NF-κB-NLRP3 network was formed and might play a key role in the pathogenesis of colitis. Given the role of LGG-EVs in the TLRs-NF-κB-NLRP3 signaling pathway, it was considered that LGG-EVs might be involved in this network to improve colitis. However, whether LGG-EVs could target TLRs and induce the synthesis of mitochondrial DNA, and activate NLRP3 signaling still needs to be further studied.

In general, the pathogenesis of colitis is closely related to the gut microbiota, although the bacteria contributing to colitis have not been determined [37]. Probiotics are emerging as functional foods to manipulate the gut microbiota and suppress the growth of pathogens [38]. However, the proposed mechanisms of probiotics on the intestinal microbiome are incomplete presently. Some studies documented that probiotics as living microorganisms belonged to the part of gut microbiota, and promoted beneficial effects when ingested in adequate amounts [39]. Other researches indicated that probiotics produced metabolic compounds and inhibited the growth of other microorganisms [40]. A recent study reported that the microbiota-derived EVs might play significant roles in host–helminth–microbiome crosstalk [41]. Therefore, it is well recognized that the probiotic-derived EVs may be an overlooked mediator in host–microbiome–probiotic interactions.

Numerous studies were focused on the effects of EVs derived from various foods on gut microbiota [42,43]. Teng et al. showed that plant-derived EVs regulated gut microbiota composition and localization, and enhanced the gut barrier function to alleviate colitis [44]. It was documented in our previous research that milk-derived EVs could alleviate colitis via modulating gut microbiota [45]. Here, we found that LGG-EVs restored the disorders of α- and β-diversity induced by DSS. Indeed, colitis has been associated with dysbiosis, defined as a significant decrease in gut microbial diversity [2]. The changes of microbial diversity were accompanied by the shift between commensal and pathogenic microorganisms, for example, the incensement of *Firmicutes*: *Bacteroidetes* (*F/B*) ratio in the DSS group. Consistently, the members of *Proteobacteria* phylum, including *Enterobacteriaceae, Escherichia_Shigella*, were commonly increased in colitis compared to the healthy and LGG-EVs treated mice, which were consistent with the results of a previous report [46]. Furthermore, LGG-EVs treatment significantly upregulated the relative abundance of *Akkermansia_muciniphila, Bifidobacterium_animalis* at the species level. Emerging evidence has indicated a close negative correlation between *Akkermansia_muciniphila* abundance and obesity, gut inflammation, or hypertension [47,48]. *Bifidobacterium_animalis*, as a generally recognized probiotic, plays a key role in gastrointestinal health and immune function [49]. In a line with the previous reports, both *Akkermansia_muciniphila* and *Bifidobacterium_animalis* were closely correlated with the valine, leucine and isoleucine degradation pathway and fluorobenzoate degradation [30,50], which were regulated upon LGG-EVs at different level. The results demonstrated that the LGG-EVs could regulate the microbial metabolic pathway via affecting the special bacteria.

It had been suggested that TLRs and NLRs could contribute to shaping the composition of the gut microbiota. Dheer et al. indicated that the expression of epithelial TLR4 shaped the microbiota and affected the functional properties of the epithelium [51]. The composition of gut microbiota and increased inflammation depends on the expression of TLR4 in colon tissues [52]. Moreover, there was lower bacteria abundance in the NLRP3^−/−^ mice colon compared to that of the WT mice. NLRP3 inflammasome is emerging to be a crucial mediator of gut homeostasis by modulating immune responses to microbiota in the gut [36]. In this study, we found that LGG-EVs could act not only as a key mediator of TLRs-NF-κB-NLRP3 but also reshape the construction and composition of gut microbiota. Namely, the abundance of *Muribaculaceae, Akkermansia, Faecalibaculum, Alistipes, Lactobcillus* had a significant negative association with pro-inflammatory cytokines, such as IL-6, TNF-α, TLR4, etc. It was demonstrated that *Akkermansia, Faecalibaculum, Alistipes* could inhibit the production of pro-inflammatory factors and play the potential beneficial effect on intestinal immune homeostasis [48,53]. However, whether LGG-EVs could mediate the TLRs-NF-κB-NLRP3 signaling pathway via regulating gut microbiota to alleviate colitis remains to be further explored.

To date, almost all research has focused on the characterization and function of EVs. In fact, EVs contain amounts of functional cargos, including RNA, proteins and etc. Further studies are required to explore which molecules play a crucial role in intestinal immune homeostasis and host health.

## 5. Conclusions

This is the first report that demonstrates that LGG-EVs can ameliorate DSS-induced colitis via mediating TLRs-NF-κB-NLRP3 signaling and reshaping gut microbiota. Oral administration of LGG-EVs might be a novel way to improve the therapeutic effect for colitis.

## Figures and Tables

**Figure 1 nutrients-13-03319-f001:**
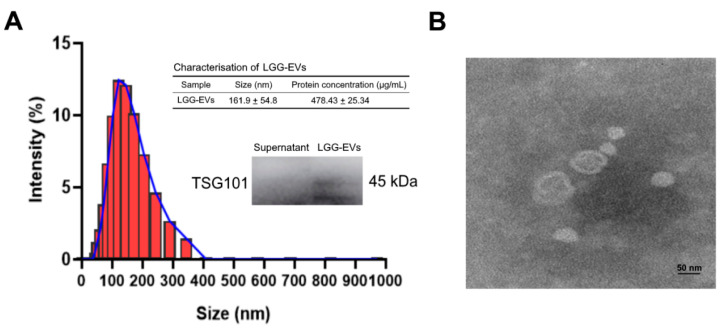
Characterization of *L. rhamnosus* GG-derived EVs. (**A**) Size distribution of LGG-EVs (*Lactobacillus rhamnosus* GG derived extracellular vesicles) was measured by DLS (Dynamic Light Scattering). The peak diameter was about 150 nm. Immunoblot bands demonstrating the presence of TSG101 (Tumor susceptibility gene 101) marker in LGG-EVs. (**B**) Transmission electron microscopy images of isolated LGG-EVs. Scale bar, 50 nm.

**Figure 2 nutrients-13-03319-f002:**
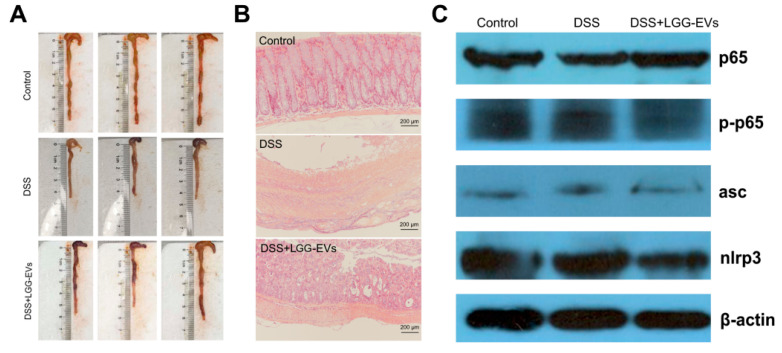
LGG-EVs alleviated DSS-induced colitis. (**A**) LGG-EVs treatment prevented the shortening of colon length in colitis mice. (**B**) Representative images of colonic sections stained with H&E for histopathologic evaluations. (**C**) Effect of LGG-EVs on NF-κB and NLRP3 signaling pathways involved in DSS-induced UC.

**Figure 3 nutrients-13-03319-f003:**
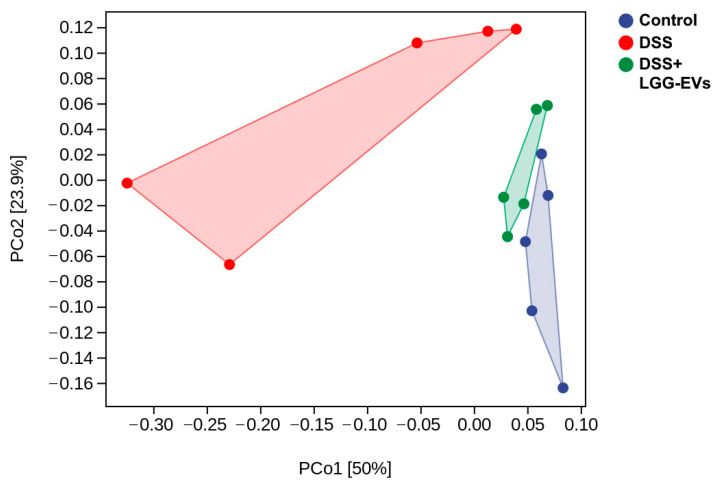
β-diversity evaluated using the weighted UniFrac-based PCoA (principal co-ordinates analysis) (*n* = 5).

**Figure 4 nutrients-13-03319-f004:**
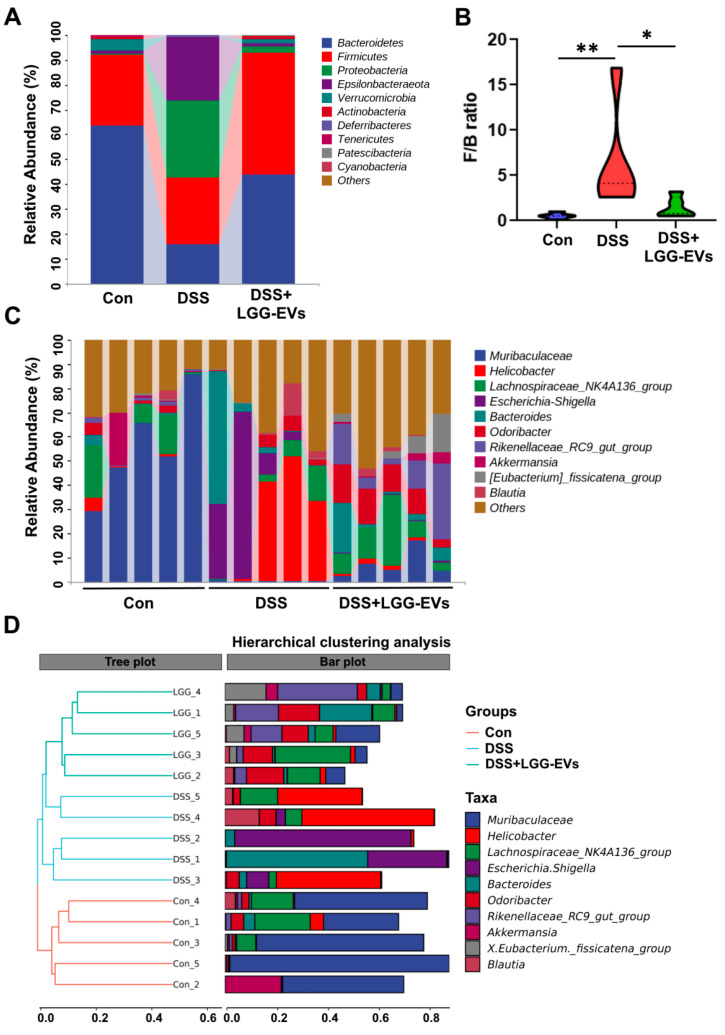
LGG-EVs reshaped the structure of gut microbiota in colitis mice. (**A**) Bar plots of the taxonomic composition at the phylum level. (**B**) The ratio of Firmicutes to Bacteroidetes in different groups. * *p* < 0.05, ** *p* < 0.01. (**C**) Bar plots of the taxonomic composition at the genus level. (**D**) Hierarchical clustering analysis based on unweighted UniFrac distance.

**Figure 5 nutrients-13-03319-f005:**
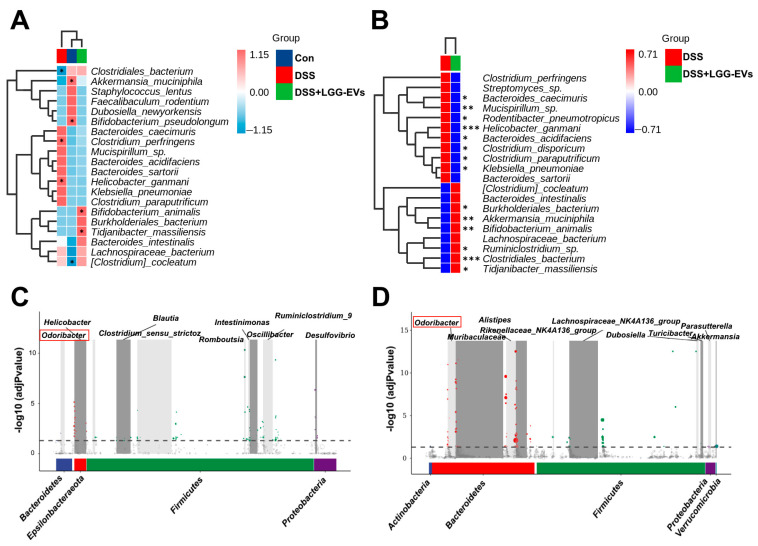
Characterization of the gut microbiota in colitis mice upon LGG-EVs treatment. Heat map showed the significant difference of characterizable bacteria among three groups (**A**) and before and after LGG-EVs treatment (**B**) at the species level. Several significantly differentially abundant taxa enriched in the DSS group (**C**) and LGG-EVs group (**D**) were identified using metagenomeSeq analysis after merging at the lowest taxonomic annotation. * *p* < 0.05, ** *p* < 0.01, *** *p* < 0.001.

**Figure 6 nutrients-13-03319-f006:**
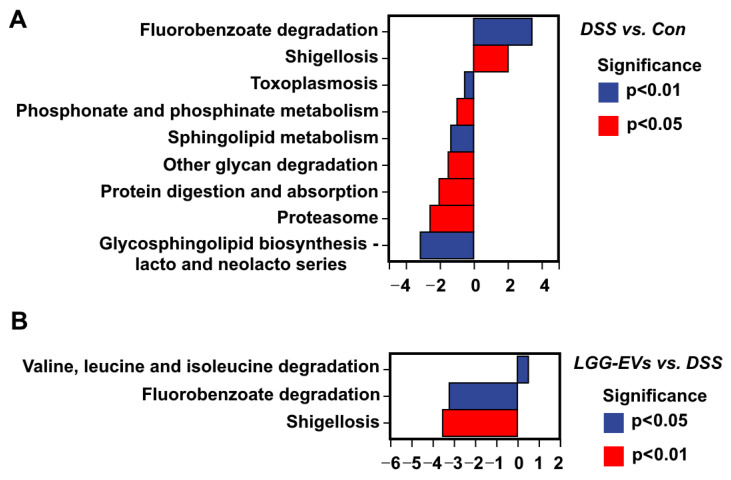
Relative abundance of KEGG pathways. Metabolism pathways of gut microbiome were predicted to show significantly different abundances in the DSS vs. Control group (**A**) and LGG-EVs vs. DSS group (**B**) according to the Kyoto Encyclopedia of Genes and Genome (KEGG) pathway analysis (*n* = 5).

**Figure 7 nutrients-13-03319-f007:**
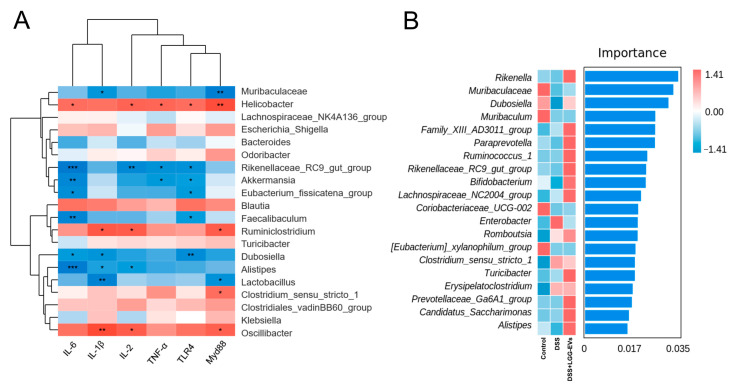
Interrelationship between gut microbiota and intestinal immune-inflammatory factors. (**A**) Heatmap of Spearman’s correlation between gut bacteria (at the genus level) and inflammatory factors. Significant difference determined at * *p <* 0.05, ** *p <* 0.01, *** *p <* 0.001. (**B**) Analysis of the bacterial profile discrepancies among three groups using random forest.

**Table 1 nutrients-13-03319-t001:** The change of body weight among different treatment groups (*n* = 7).

Groups/BW (g)	−6 Days	−4 Days	−2 Days	0 Days	2 Days	4 Days	6 Days	8 Days
Control	21.58 ± 0.69	22.29 ± 0.90	22.38 ± 1.18	23.04 ± 1.11	24.33 ± 1.25	23.94 ± 1.54	23.45 ± 1.56 **	23.31 ± 1.20 **
DSS	21.60 ± 0.76	22.55 ± 0.85	22.72 ± 1.31	23.28 ± 1.49	24.17 ± 1.84	21.93 ± 1.22	20.02 ± 0.94	18.03 ± 0.66
DSS + LGG-EVs	21.03 ± 0.80	21.37 ± 1.03	21.48 ± 1.12	23.19 ± 1.51	23.91 ± 1.64	22.74 ± 1.66	21.58 ± 1.94	20.33 ± 1.68 *

The body weight (BW) was measured 6 days before DSS (Dextran Sulfate Sodium)-induced colitis. * *p* < 0.05, ** *p* < 0.01 vs. DSS group. LGG-EVs, *Lactobacillus rhamnosus* GG derived extracellular vesicles. Data were presented by mean ± SD (Standard Deviation). D’Agostino and Pearson test were used to verify normality of distribution of data.

**Table 2 nutrients-13-03319-t002:** The organ index and colon length were evaluated in different groups (*n* = 7).

Groups/Symptoms	Spleen Index (g)	Liver Index (g)	Colon Length (cm)
Control	2.35 ± 0.41 ^a^	3.80 ± 0.30 ^a^	6.21 ± 0.66 ^c^
DSS	4.48 ± 0.95 ^c^	3.73 ± 0.26 ^a^	4.55 ± 0.34 ^a^
DSS + LGG-EVs	3.68 ± 0.22 ^b^	3.72 ± 0.17 ^a^	5.53 ± 0.49 ^b^

^a–c^ Significant (*p* < 0.05) difference between samples. D’Agostino and Pearson test were used to verify normality of distribution of data.

**Table 3 nutrients-13-03319-t003:** LGG-EVs inhibited DSS-induced the up-regulation of related inflammatory genes.

Genes/Groups	Control	DSS	DSS + LGG-EVs
IL-6	1.02 ± 0.11 *	1.27 ± 0.12	0.87 ± 0.07 **
IL-1β	0.91 ± 0.09 ***	2.25 ± 0.14	1.08 ± 0.17 ***
IL-2	1.03 ± 0.06 ***	3.12 ± 0.56	1.13 ± 0.59 ***
TNF-α	1.00 ± 0.03 ***	2.18 ± 0.17	1.12 ± 0.32 ***
TLR-4	1.12 ± 0.21 ***	3.08 ± 1.07	0.90 ± 0.29 ***
Myd88	1.13 ± 0.16 ***	1.77 ± 0.15	1.30 ± 0.14 **

Data were presented as mean ± SD (*n* = 3). * *p* < 0.05, ** *p* < 0.01 and *** *p* < 0.001 vs. DSS group. D’Agostino and Pearson test were used to verify normality of distribution of data.

**Table 4 nutrients-13-03319-t004:** Expression level of proteins in NF-κB and NLRP3 signaling pathways upon LGG-EVs treatments.

Proteins/Groups	Control	DSS	DSS + LGG-EVs
p-p65/p65	0.81 ± 0.03 ***	1.11 ± 0.06	0.55 ± 0.02 ***
ASC	0.07 ± 0.00 **	0.08 ± 0.01	0.07 ± 0.01 *
NLRP3	0.46 ± 0.03 *	0.54 ± 0.03	0.48 ± 0.04 (p = 0.058)

Normalized to β-actin. Data were presented as mean ± SD (*n* = 3). * *p* < 0.05, ** *p* < 0.01 and *** *p* < 0.001 vs. DSS group. D’Agostino and Pearson test were used to verify normality of distribution of data.

**Table 5 nutrients-13-03319-t005:** Alpha diversity indexes calculated with QIIME2 according to ASV/OTU numbers of each group (*n* = 5).

Alpha Diversity Indexes/Groups	Control	DSS	DSS + LGG-EVs
Chao 1	3856.50 ± 1004.11 ***	869.44 ± 393.80	2072.55 ± 510.40 *
Shannon	8.90 ± 1.37 ***	4.74 ± 0.50	7.43 ± 0.84 *
Observed_species	3381.78 ± 1095.07 ***	707.50 ± 332.89	1711.36 ± 451.48
Faith_pd	134.42 ± 24.83 ***	54.67 ± 16.30	106.05 ± 17.24 **
Simpon	0.98 ± 0.03 ***	0.83 ± 0.06	0.96 ± 0.03 ***
Pielou_e	0.76 ± 0.09 ***	0.50 ± 0.02	0.69 ± 0.06 **

Data were presented as the mean ± SD. * *p* < 0.05, ** *p* < 0.01 and *** *p* < 0.001 vs. DSS group. D’Agostino and Pearson test were used to verify normality of distribution of data.

**Table 6 nutrients-13-03319-t006:** Relative abundance of bacteria of classical producer for short-chain fatty acids at the family level (*n* = 5).

Family/Groups	Control (%)	DSS (%)	DSS + LGG-EVs (%)
Lachnospiraceae	0.15 ± 0.15	0.13 ± 0.14	0.30 ± 0.09 *
Ruminococcaceae	0.03 ± 0.02	0.08 ± 0.08	0.08 ± 0.07
Lactobacillaceae	(0.01, 0.00, 0.04) ^a^	(0.01, 0.00, 0.02) ^a^	(0.01, 0.00, 0.02) ^a^
Clostridiaceae_1	0.00 ± 0.00	0.01 ± 0.01 *	0.01 ± 0.00 *
Clostridiales_vadinBB60	(0.00, 0.00, 0.01) ^a^	(0.00, 0.00, 0.01) ^a^	(0.01, 0.00, 0.03) ^a^ *

Data accord with normal distribution were presented as the mean ± SD. * *p* < 0.05, Control vs. DSS vs. DSS + LGG-EVs. D’Agostino and Pearson test were used to verify normality of distribution of data. ^a^ Data were presented by median, min and max values.

**Table 7 nutrients-13-03319-t007:** Relative abundance of bacteria classical producer for short-chain fatty acids at the genus level (*n* = 5).

Genus/Groups	Control (%)	DSS (%)	DSS + LGG-EVs (%)
Lachnospiraceae_NK4A136	0.09 ± 0.09	0.05 ± 0.06	0.12 ± 0.10
Ruminiclostridium_9	(0.00, 0.00, 0.01) ^a^	(0.00, 0.00, 0.01) ^a^	(0.02, 0.01, 0.04) ^a^ *
Lactobacillus	0.02 ± 0.02	0.00 ± 0.00	0.01 ± 0.01
Clostridium_sensu_stricto_1	0.00 ± 0.00	0.01 ± 0.01 *	0.01 ± 0.00 *
Clostridiales_vadinBB60	(0.00, 0.00, 0.01) ^a^	(0.00, 0.00, 0.01) ^a^	(0.01, 0.00, 0.03) ^a^
Faecalibaculum	(0.00, 0.00, 0.00) ^a^	(0.00, 0.00, 0.00) ^a^	(0.00, 0.00, 0.01) ^a^ *

Data accord with normal distribution were presented as the mean ± SD. * *p* < 0.05, Control vs. DSS vs. DSS + LGG-EVs. D’Agostino and Pearson test were used to verify normality of distribution of data. ^a^ Data were presented by median, min and max values.

## Data Availability

The data presented in this study are available on request from the corresponding author.

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
