# Peer review of "Lactobacillus rhamnosus GG Derived Extracellular Vesicles Modulate Gut Microbiota and Attenuate Inflammatory in DSS-Induced Colitis Mice"

_nutrients, 2021, doi:10.3390/nu13103319_

Round 1

Reviewer 1 Report

Review of “Lactobacillus rhamnosus GG derived extracellular vesicles modulate gut microbiota and attenuate inflammatory in DSS-induced colitis mice” (nutrients-1381776)

This study investigated the protective effect of probiotics LGG-EV on DSS-induced colitis and its mechanism. This study is well written, and this reviewer has a few comments.

  1. In Discussion section (Line 386-388), “Consistently, the members of Proteobac teria phylum, including Enterobacteriaceae, Escherichia_Shigella (Figure S2), were commonly increased in colitis compared to the healthy and L-EVs treated mice,”. To add these results including Figure 2S in the Results section and added the methos of LEfSe in Methods section.
  2. Please add the explanation of the association between change of gut microbiota and inflammation in the Discussion section.
  3. The resolutions of Figure 3 E and F are low.

Author Response

Dear editor,

Thank you for your consideration for our manuscript (ID: nutrients-1381776) entitled “Lactobacillus rhamnosus GG derived extracellular vesicles modulate gut microbiota and attenuate inflammatory in DSS-induced colitis mice”, by Tong et al. We are thankful to the reviewers and the editor for pointing out some important modifications needed in the report. We have thoughtfully taken into account these comments. The response to the reviewers’ concerns is given point by point as following.

Response to the reviewers’ comments

Reviewer 1

Comment 1: In Discussion section (Line 386-388), “Consistently, the members of Proteobac teria phylum, including Enterobacteriaceae, Escherichia, Shigella (Figure S2), were commonly increased in colitis compared to the healthy and L-EVs treated mice,”. To add these results including Figure 2S in the Results section and added the methos of LEfSe in Methods section.

Response: Thank you for your advice. The relative results for Figure S2 have added to the section of Results (Please see Line 253-255). The methods of LEfSe have been supplemented in the section of Methods (Please see Line 144).

Comment 2: Please add the explanation of the association between change of gut microbiota and inflammation in the Discussion section.

Response: The association between change of gut microbiota and inflammation was supplemented. “Namely, the abundance of Muribaculaceae, Akkermansia, Faecalibaculum, Alistipes, Lactobcillus etc had significant negative association with pro-inflammatory cytokines, such as IL-6, TNF-α, TLR4 etc. It had been demonstrated that Akkermansia, Faecalibacu-lum, Alistipes etc could inhabit the production of pro-inflammatory factors and play the potential beneficial effect on intestinal immune homeostasis [49,54]. However, whether LGG-EVs could mediated TLRs-NF-κB-NLRP3 signaling pathway via regulating gut microbiota to alleviate colitis remains to be further explored.” (Line 445-451)

Comment 3: The resolutions of Figure 3 E and F are low.

Response: Thanks for your suggestion. The higher resolution Figure have added in manuscript. (Please see Line 325)

Reviewer 2 Report

The manuscript entitled „ Lactobacillus rhamnosus GG derived extracellular vesicles modulate gut microbiota and attenuate inflammatory in DSS-induced colitis mice” presents interesting issue but some problems must be corrected.

Major:

  1. Authors should avoid too strong and too general statements which are not based on the literature, e.g. “Probiotics play a crucial role in the prevention of UC onset and relapse” (there are numerous factors which influence, so we can not state that one of them plays a “crucial” role); “These findings revealed the new mechanisms of LGG in attenuating inflammatory mediated by extracellular vesicles” (based on the presented study we can not state that it “revealed the new mechanisms”, but rather that it proposed some novel areas)
  2. While presenting the issues associated with probiotics Authors must properly reflect the current state of knowledge, as the presented version is misleading. Authors should be aware that only some strains have probiotic properties, so Authors can not state “probiotics […], such as Bifidobacterium infantis”, as only some strains of Bifidobacterium infantis, have probiotic properties, so Authors must indicate those strains which have those properties. I suppose that they mean B. infantis 35624, but it must be clearly stated. Similarly, Authors should clearly define “Lactobacillus GG” as Lactobacillus rhamnosus GG. All the other issues associated with the proper presentation of data associated with probiotics should be corrected in the whole manuscript.

Abstract:

Authors should precisely define the aim of the study, e.g. “the aim of the study was…” instead of just indicating what was done within the study

Authors should present specific numeric results accompanied by the results of their statistical analysis (p-Value)

Introduction:

Authors should broaden the presented information to reflect properly the perspective to include following issues:

  • Potential influence of the other probiotic strains – Authors should refer specific studies which confirmed this influence
  • Influence of prebiotics – Authors should refer specific studies which confirmed this influence and describe which prebiotics may be used

Materials and Methods:

“standard diet” should be described and the basic nutritional value should be presented

The housing conditions should be described, while Authors should indicate how did they try to reduce the stress associated with an experiment

It seems that Authors did not verify the normality of distribution and they treated all the variables as normally distributed, but it may be supposed that data for some variables were characterised by the distribution different than normal.

Authors should (1) verify the normality of distribution, (2) for normally distributed data present mean and SD values, but for the other distributions – present median, min and max values, (3) apply adequate statistical tests, that are based on the distribution.

Results:

For normally distributed data Authors should present mean and SD values, but for the other distributions – present median, min and max values.

Authors should apply adequate statistical tests, that are based on the distribution.

Some figures are extremely hard to follow, so they should be replaced by tables, e.g. Figure 2 a, b, c, f, g, h, i, k, l, 3 a, 5 a, b

Discussion:

Authors can not reproduce results in this section. Authors should include

Authors should: (1) compare gathered data with the results by other authors, (2) formulate implications of the results of their study and studies by other authors, (3) formulate the future areas which should be studied.

Authors should indicate the limitations of their study

Conclusions:

Authors should briefly formulate the conclusions that are directly associated with their study (conclusion such as “Oral administration of L-EVs might be a novel way to improve therapeutic effect for colitis.” May not be formulated based on the presented analysis)

Authors Contribution:

Manuscript preparation should be reflected here.

Author Response

Dear editor,

Thank you for your consideration for our manuscript (ID: nutrients-1381776) entitled “Lactobacillus rhamnosus GG derived extracellular vesicles modulate gut microbiota and attenuate inflammatory in DSS-induced colitis mice”, by Tong et al. We are thankful to the reviewers and the editor for pointing out some important modifications needed in the report. We have thoughtfully taken into account these comments. The response to the reviewers’ concerns is given point by point as following.

Response to the reviewers’ comments

Comment 1: Authors should avoid too strong and too general statements which are not based on the literature, e.g. “Probiotics play a crucial role in the prevention of UC onset and relapse” (there are numerous factors which influence, so we can not state that one of them plays a “crucial” role); “These findings revealed the new mechanisms of LGG in attenuating inflammatory mediated by extracellular vesicles” (based on the presented study we can not state that it “revealed the new mechanisms”, but rather that it proposed some novel areas)

Response: Thank you for your advice. We have revised the relative contents according to your suggestion. “Probiotics play a crucial role in the prevention of UC onset and relapse…” was changed to “Probiotics have the potential beneficial effect in the prevention of UC…”; “These findings revealed the new mechanisms of LGG…” was revised to “These findings proposed a novel perspective of L. rhamnosus GG…”. (Please see Line 14 and Line 25)

Comment 2: While presenting the issues associated with probiotics Authors must properly reflect the current state of knowledge, as the presented version is misleading. Authors should be aware that only some strains have probiotic properties, so Authors can’t state “probiotics […], such as Bifidobacterium infantis”, as only some strains of Bifidobacterium infantis, have probiotic properties, so Authors must indicate those strains which have those properties. I suppose that they mean B. infantis 35624, but it must be clearly stated. Similarly, Authors should clearly define “Lactobacillus GG” as Lactobacillus rhamnosus GG. All the other issues associated with the proper presentation of data associated with probiotics should be corrected in the whole manuscript.

Response: Thanks for your consideration. We have revised the contents of the special strains. “It was reported that Bifidobacterium infantis 35624 could exert beneficial immunoregulatory effects in the mucosal immune system [3]. Lactobacillus rhamnosus GG could enhance intestinal functional maturation and IgA production and protect against colitis [4]. Lactobacillus bulgaricus attenuated the clinical signs of intestinal inflammation inducing a decrease of inflammatory cytokines [5].” (Please see Line 41-45); In addition, “Lactobacillus GG” was clearly defined as Lactobacillus rhamnosus GG.

Comment 3:

Abstract:

Authors should precisely define the aim of the study, e.g. “the aim of the study was…” instead of just indicating what was done within the study

Authors should present specific numeric results accompanied by the results of their statistical analysis (p-Value)

Response: Thanks for your advice. The aim of the study has been added in the section of Abstract. “The aim of this study is to explore the effect of extracellular vesicles released from L. rhamnosus GG (LGG-EVs) on dextran sulfate sodium (DSS)-induced colitis, and propose the underlying mechanism of LGG-EVs for protecting against colitis” (Line 17-19). Moreover, the specific numeric results including P-value have been presented. please see Line 20 and 23.

Comment 4:

Introduction:

Authors should broaden the presented information to reflect properly the perspective to include following issues:

Potential influence of the other probiotic strains – Authors should refer specific studies which confirmed this influence

Influence of prebiotics – Authors should refer specific studies which confirmed this influence and describe which prebiotics may be used

Response: Thanks for your advice. The potential influence and effect of the special probiotics or prebiotics in the colitis have been supplemented into Introduction, “It was reported that Bifidobacterium infantis 35624 could exert beneficial immunoregulatory effects in the mucosal immune system [3]. Lactobacillus rhamnosus GG (L. rham-nosus GG) could enhance intestinal functional maturation and IgA production and protect against colitis [4]. Lactobacillus bulgaricus attenuated the clinical signs of intestinal inflammation inducing a decrease of inflammatory cytokines [5]. In addition, prebiotics might change the composition of gut microbiota, improve the function of the intestinal barrier, enhance intestinal immunity. It was shown that prebiotic fructans and resveratrol treatment could increase the amount of Bifidobactrium and Lactobacillus in the DSS-induced colitis [6]. Therefore, prebiotics and probiotics may represent a valid armamentarium to alleviate colitis, while the mechanism of the action is still unclear.” (Line 41-50)

Comment 5:

Materials and Methods:

  1. “standard diet” should be described and the basic nutritional value should be presented
  2. The housing conditions should be described, while Authors should indicate how did they try to reduce the stress associated with an experiment
  3. It seems that Authors did not verify the normality of distribution and they treated all the variables as normally distributed, but it may be supposed that data for some variables were characterised by the distribution different than normal. Authors should (1) verify the normality of distribution, (2) for normally distributed data present mean and SD values, but for the other distributions – present median, min and max values, (3) apply adequate statistical tests, that are based on the distribution.

Response: Thanks for your suggestion.

  1. The basic nutritional components have been described and added into Table S1 in Supplemental_Text. Please see it in Supplemental.
  2. The housing conditions and how to reduce the stress associated with the experiment have been clarified in the section of Method, “The relative humidity was 30 % to 70 % in the experimental room. The bedding-change, cage-washing frequently and the preparation of recycled air were used to reduce the stress associated with the experiment.” Please Line 92-94.
  3. Thank you for your suggestion. We have verified the normality of distribution via SPSS 22 (Please see all the footer of Tables) and presented data using mean and SD values in all the Table of manuscript. For the other distributions, median, min and max values were presented in Figure 2. The One-way analysis of variance (ANOVA), Duncan test and Normality test have been used to analyze the data.

Comment 6:

Results:

  1. For normally distributed data Authors should present mean and SD values, but for the other distributions – present median, min and max values. Authors should apply adequate statistical tests, that are based on the distribution.
  2. Some figures are extremely hard to follow, so they should be replaced by tables, e.g. Figure 2 a, b, c, f, g, h, i, k, l, 3 a, 5 a, b

Response: Thank you for your advice.

  1. We have verified the normality of distribution and presented data using mean and SD values in all the Table of manuscript. For the other distributions, median, min and max values were presented in Figure 2. The One-way analysis of variance (ANOVA), Duncan test and Normality test have been used to analyze the data.
  2. Some Figures were replaced by Tables according to your suggestion. Please see Table 1, Table 2, Table 3, Table 4, Table 5, and Table S3 in manuscript and supplemental text.

Comment 7:

Discussion:

  1. Authors can not reproduce results in this section. Authors should: (1) compare gathered data with the results by other authors, (2) formulate implications of the results of their study and studies by other authors, (3) formulate the future areas which should be studied.
  2. Authors should indicate the limitations of their study

Response: Thank you for your advice.

  1. According to your suggestion, we have added some discussions compared to other results and the future areas which should be studied in manuscript. Please see Line 446-450, 450-452.
  2. The limitations of our study and the future study areas have been supplemented in Discussion section, “To date, almost researches focused on the characterization and function of EVs. In fact, EVs contain significant amounts of functional cargos, including RNA, proteins and etc. Further studies are required to explore which molecules plays a crucial role in the intestinal immune homeostasis and host healthy in the future.”. please see 454-457.

Comment 8:

Conclusions:

Authors should briefly formulate the conclusions that are directly associated with their study (conclusion such as “Oral administration of LGG-EVs might be a novel way to improve therapeutic effect for colitis.” May not be formulated based on the presented analysis)

Response: Thanks for your advice. The section of conclusions was revised. Please see Line 460-464.

Comment 9: Authors Contribution: Manuscript preparation should be reflected here.

Response: Thanks for your advice. Authors Contribution was added. Please see Line 470.

Round 2

Reviewer 2 Report

The manuscript entitled „ Lactobacillus rhamnosus GG derived extracellular vesicles modulate gut microbiota and attenuate inflammatory in DSS-induced colitis mice” presents interesting issue but some problems must be corrected.

Major:

While presenting the issues associated with probiotics Authors must properly reflect the current state of knowledge, as the presented version is misleading. Authors should be aware that only some strains have probiotic properties, so Authors can not state “Lactobacillus bulgaricus attenuated the clinical signs of intestinal inflammation inducing a decrease of inflammatory cytokines”, as only some strains, have probiotic properties, so Authors must indicate those strains which have those properties. All the other issues associated with the proper presentation of data associated with probiotics should be corrected in the whole manuscript.

Materials and Methods:

“standard diet” should be described and the basic nutritional value should be presented – Authors should not present only components (corn, soybean meal, etc.), but above all the nutritional value (fat, protein, etc.)

It still seems that Authors did not verify the normality of distribution and they treated all the variables as normally distributed, but it may be supposed that data for some variables were characterised by the distribution different than normal. Authors should indicate what normality test did they use and what were the results of verification.

For normally distributed data present mean and SD values, but for the other distributions – present median, min and max values

Authors should apply adequate statistical tests, that are based on the distribution.

Results:

For normally distributed data Authors should present mean and SD values, but for the other distributions – present median, min and max values.

Authors should apply adequate statistical tests, that are based on the distribution.

Some figures are extremely hard to follow, so they should be replaced by tables, e.g. Figure 2 c, e, f

Discussion:

Authors should indicate the limitations of their study – all of them

Authors Contribution:

It seems that contribution of majority of Authors was only minor as they did not participate in preparing manuscript. There is a serious risk of a guest authorship procedure which is forbidden. In such case they should be rather presented in Acknowledgements Section and not be indicated as authors of the study.

Author Response

Dear reviewer,

Because there are the morjority of figures in the point-by-point response file, which couldn't be presented in this Text. Please see the uploaded dodument. Thank you very much!

Yours sincerely,

Prof. Huaxi Yi
